# Proteomic Analyses Reveal the Role of Alpha-2-Macroglobulin in Canine Osteosarcoma Cell Migration

**DOI:** 10.3390/ijms25073989

**Published:** 2024-04-03

**Authors:** Sylwia S. Wilk, Katarzyna Michalak, Ewelina P. Owczarek, Stanisław Winiarczyk, Katarzyna A. Zabielska-Koczywąs

**Affiliations:** 1Department of Small Animal Diseases and Clinic, Institute of Veterinary Medicine, Warsaw University of Life Sciences, Nowoursynowska 159c, 02-787 Warsaw, Poland; sylwia_wilk@sggw.edu.pl (S.S.W.); eowczarek@iimcb.gov.pl (E.P.O.); 2Department of Epizootiology and Clinic of Infectious Diseases, University of Life Sciences, Głęboka 30, 20-612 Lublin, Poland; katarzyna.michalak@up.lublin.pl (K.M.); stanislaw.winiarczyk@up.lublin.pl (S.W.); 3Laboratory of RNA Biology, International Institute of Molecular and Cell Biology in Warsaw, 4 Ks. Trojdena, 02-109 Warsaw, Poland; 4National Veterinary Research Institute, Aleja Partyzantów 5, 24-100 Puławy, Poland

**Keywords:** A2M, canine osteosarcoma, MALDI-TOF/TOF MS, Simple Western, wound-healing assay

## Abstract

Canine osteosarcoma (OSA) is an aggressive bone neoplasia with high metastatic potential. Metastasis is the main cause of death associated with OSA, and there is no current treatment available for metastatic disease. Proteomic analyses, including matrix-assisted laser desorption/ionisation-time of flight mass spectrometry (MALDI TOF/TOF MS), are widely used to select molecular targets and identify proteins that may play a key role in primary tumours and at various steps of the metastatic cascade. The main aim of this study was to identify proteins differently expressed in canine OSA cell lines with different malignancy phenotypes (OSCA-8 and OSCA-32) compared to canine osteoblasts (CnOb). The intermediate aim of the study was to compare canine OSA cell migration capacity and assess its correlation with the malignancy phenotypes of each cell line. Using MALDI-TOF/TOF MS analyses, we identified eight proteins that were significantly differentially expressed (*p* ≤ 0.05) in canine OSA cell lines compared to CnOb: cilia- and flagella-associated protein 298 (CFAP298), general transcription factor II-I (GTF2I), mirror-image polydactyly gene 1 protein (MIPOL1), alpha-2 macroglobulin (A2M), phosphoglycerate mutase 1 (PGAM1), ubiquitin (UB2L6), ectodysplasin-A receptor-associated adapter protein (EDARADD), and leucine-rich-repeat-containing protein 72 (LRRC72). Using the Simple Western technique, we confirmed high A2M expression in CnOb compared to OSCA-8 and OSCA-32 cell lines (with intermediate and low A2M expression, respectively). Then, we confirmed the role of A2M in cancer cell migration by demonstrating significantly inhibited OSA cell migration by treatment with A2M (both at 10 and 30 mM concentrations after 12 and 24 h) in a wound-healing assay. This study may be the first report indicating A2M’s role in OSA cell metastasis; however, further in vitro and in vivo studies are needed to confirm its possible role as an anti-metastatic agent in this malignancy.

## 1. Introduction

Osteosarcoma (OSA) is a mesenchymal-derived malignancy and the most common primary bone neoplasia in dogs [1]. The tumour is highly aggressive, and most affected animals die due to lung metastasis within 3–12 months of diagnosis, despite treatment [2]. Currently, methods of canine OSA therapy include surgery (either limb amputation or limb-sparing surgery), radiotherapy, and chemotherapy [3]. Currently, there are no available methods for OSA metastasis treatment. The process of metastasis is complex and consists of five main stages: (1) local invasion and migration, (2) intravasation, (3) survival during circulation, (4) extravasation, and (5) outgrowth at a secondary site [4]. Knowledge of molecular targets for metastasis in OSA may enhance future treatment strategies. However, little is known about the genes and proteins involved in each step of the metastatic cascade. Evidence has been shown for the involvement of the epithelial growth factor receptor (EGFR) in neoplastic cell migration; hepatocyte growth factor/scatter factor (HGF/SF) and c-Met-receptor for invasion; and ΔNp63, ezrin, phosphorylated ezrin–radixin–moesin (p-ERM), Snail2, miR-9, and miR-34a for both migration and invasion [5]. However, the correlation between mRNA levels and protein concentrations remains typically poor and may be associated with post-transcriptional, translational, and degradation regulation and their role in determining protein concentrations [6,7]. Consequently, genome-wide expression studies should be followed by analyses of particular protein products using direct proteomics techniques, including mass spectrometry [7].

Matrix-assisted laser desorption/ionisation-time of flight spectrometry (MALDI-TOF/TOF MS) is applied to the analysis of biomolecules including carbohydrates, DNA chains, peptides, and proteins [8]. The method has been widely used in biology, especially in identifying microorganisms [9]. This technique’s utility in canine and feline oncological studies has been reported for canine mammary carcinomas and feline fibrosarcomas [10,11]. Klopfleisch et al. [10] identified 21 proteins that were significantly differentially expressed in metastatic and non-metastatic canine mammary carcinomas. Most of these proteins are associated with biological functions linked to metastatic spread such as cell adhesion, extracellular matrix modulation, and hypoxia resistance, making them potentially new markers for metastatic mammary carcinomas [10]. Zabielska-Koczywąs et al. [11] identified three significantly (*p* ≤ 0.05) differentially expressed proteins (annexin A5 (ANXA5), annexin A3 (ANXA3), and meiosis-specific nuclear structural protein 1 (MNS1)) in doxorubicin-resistant fibrosarcomas compared to doxorubicin-sensitive ones, and claimed that they may be involved in the chemotherapy resistance of feline fibrosarcomas. Our main aim for this study was to identify proteins that are differentially expressed in canine OSA cell lines (OSCA-8 and OSCA-32) with different phenotypes of aggressiveness, and CnOb. We also sought to validate the roles of selected proteins in cancer cell migration, an important step in the metastatic cascade. Our intermediate aim for this study was to compare OSCA-8 and OSCA-32 cell migration capacity and assess their correlation with malignancy phenotypes from each cell line. Our comprehensive proteomic analyses of OSA cells with different molecular phenotypes of aggressiveness expanded our knowledge of tumour biology and could be a first step toward targeted therapy [12]. To the best of our knowledge, we are the first to perform proteomic analyses of canine OSA cell lines and CnOb using MALDI TOF/TOF MS analyses.

## 2. Results

### 2.1. Proteins Differently Expressed in Canine OSA Cells in Comparison to Osteoblasts

Eight out of two thousand protein spots detected in the process of two-dimensional electrophoresis (2DE) were selected and identified using MALDI TOF/TOF MS analyses as significantly differentially expressed (*p* ≤ 0.05) in canine OSA cell lines (OSCA-8 and OSCA-32) compared to CnOb (Table 1). Two of these, cilia- and flagella-associated protein 298 (CFAP298) and general transcription factor II-I (GTF2I), were upregulated in OSA cell lines, while six proteins, namely mirror-image polydactyly 1 protein (MIPOL1), alpha-2-macroglobulin (A2M), phosphoglycerate mutase 1 (PGAM1), ubiquitin (UB2L6), ectodysplasin-A receptor-associated adapter protein (EDARADD), and leucine-rich-repeat-containing protein 72 (LRRC72), were downregulated in OSA cells compared to the control (Figure 1 and Figure 2). Figure 1 shows an example of polyacrylamide gel with marked spots. The top-scoring results were obtained from species including humans (*Homo sapiens*), mice (*Mus musculus*), and domestic cattle (*Bos taurus*). These results can be explained by the limited availability of dog sequences in the Swiss-Prot database used in our research (www.uniprot.org, accessed on 3 December 2020).

### 2.2. Expression of A2M Protein in Canine OSA Cell Lines and CnOb

Primary antibody validation for the Simple Western technique, including antibody saturation and protein linear range assessment, is essential for performing protein expression analyses. The primary A2M antibody MAB1938 (Bio-Techne, Minneapolis, MN, USA) saturation reached a minimum of 90% using a dilution of 1:6.25. The protein linear range was assessed for all samples (OSCA-8, OSCA-32, and CnOb) with a 1:6.25 antibody concentration between 0.2 and 0.5 mg/mL (Appendix A). The Simple Western analyses showed high A2M expression in CnOb, and low and intermediate A2M expression in the OSCA-32 and OSCA-8 cell lines, respectively (Figure 3). The positive control for A2M expression is shown in Appendix A.

### 2.3. The Influence of A2M on the Migration Rate of OSCA-8 and OSCA-32 Cell Lines

An in vitro wound-healing assay was used to analyse the A2M protein’s inhibition of OSCA-8 and OSCA-32 cell migration. In the cell lines, both the applied concentrations (10 mM and 30 mM) of A2M significantly (*p* ≤ 0.001 after 12 h and *p* ≤ 0.001 after 24 h) inhibited cancer cell migration compared to the control (without A2M) (Figure 4). Additionally, the wound-healing assay evaluated the migration rate of OSCA-8 (highly malignant phenotype of OSA cells) and OSCA-32 (intermediately malignant phenotype of OSA cells) compared to CnOb (Figure 5, Figure 6 and Figure 7). The migration rate of OSA cells was significantly higher (*p* ≤ 0.001) than of the control at t = 12 h and t = 24 h (Figure 5). Moreover, the migration rate was significantly higher (*p* ≤ 0.001) in highly malignant OSCA-8 cells than in intermediately malignant OSCA-32 cells at t = 12 h (Figure 5).

## 3. Discussion

MALDI-TOF/TOF MS is a widely used technique in proteomics and metabolomics with practical applications, especially in microbiology [9], for identifying bacteria and fungi [13,14]. However, MALDI-TOF/TOF MS has also been used in human oncological research as a potential diagnostic tool for identifying tumour biomarkers, including head and neck squamous cell carcinoma, prostate cancer, renal cancer, bladder cancer, lung cancer, and OSA [15,16,17,18]. In veterinary oncology, there are reports of using this method in the proteomic analysis of several tumours including mammary cancer in dogs [10], various types of canine oral tumours (including melanoma, squamous cell carcinoma, and benign tumours) [19], canine and feline lymphoma [20,21], feline fibrosarcoma [11], and canine OSA [12]. The advantages of using MALDI TOF/TOF MS in oncological studies include the high sensitivity and speed of analyses, wide mass range coverage, and the low cost of consumables. Additionally, the method is considered easy to use and requires a low volume of samples for analyses [22]. However, the crucial disadvantage of the technique is its dependence on data quality and instrument settings, including the calibration used for peak identification and its protocols [23]. The cost of the MALDI TOF/TOF MS instrument is also not negligible, nor is the need for an experienced person to perform the analyses. Moreover, the software version used to generate and visualise MS data may influence its accuracy. The row data of MS analyses typically contain noise sources among the true signal elements, and pre-processing is usually needed before final analysis [22]. As a result, the data obtained by MALDI TOF/TOF MS analyses should be further validated with other techniques such as Western blot or Simple Western, i.e., quantitative hands-off capillary-based protein separation and immunodetection methods. Simple Western has advantages over traditional Western blot, such as full automation (which decreases the occurrence of human errors during the analyses), reproducibility, ease of use, and the quantification of protein expression [24]. In our studies, MALDI-TOF/TOF MS analysis revealed different expressions in eight proteins of OSA cells compared to CnOb (Figure 2, Table 1), out of which the qualitative Simple Western technique further confirmed the lower expression of A2M in canine OSA cells compared to osteoblasts (Figure 3).

Alpha-2-macroglobulin (A2M) is a plasma protein that modulates the activity of proteases by inhibiting them [25]. A2M protein has the molecular structure of tetramers, and its identical subunits have a molecular weight of approximately 179,000 Da, including the carbohydrate groups [26,27]. For human samples in the Simple Western method, a specific band was detected for A2M at approximately 178 kDa [Simple Western^TM^ producer guidelines: https://www.rndsystems.com/products/human-alpha-2-macroglobulin-antibody-257316_mab1938#product-citations, accessed on 15 March 2021]. The main characteristic region of A2M proteins is called the “bait region”, which contains cleavage sites for different proteinases. However, A2M may exhibit differences in molecular weight, depending on heating and reduced conditions during in vitro procedures [28]. The mass of A2M may vary according to specific post-translational modifications such as glycosylation or disulphide bond creation.

A2M modulates cell proliferation and may function as a hormone, immune modulator, and cytokine [25]. Recently, A2M has been shown to influence tumour cell adhesion, migration, and growth by inhibiting tumour-promoting signalling pathways, e.g., PI3K/AKT and SMAD, and upregulating *PTEN* [29]. Lindner et al. [30] described human astrocytoma cells’ dependence on migration and invasion upon exposure to A2M in serial concentrations, where a negative correlation between the A2M concentration and migration and invasion rates was observed. According to the authors, the malignant properties of the tumour can be inhibited by acting on the low-density lipoprotein receptor-related protein 1 (LRP1), which is a receptor for A2M. A probable mechanism for A2M cancer cell migration inhibition in astrocytoma cells may be related to impeding the Wnt/beta-catenin signalling pathway [30].

Moreover, A2M-inactivated proteolytic enzymes, including plasmin, urokinase-type plasminogen activators, and metalloproteases, have demonstrated involvement in the tumour invasion process [31,32,33]. Additionally, Amini et al. [34] established *A2M* expression in hibernating common carp plasma (HCCP) using real-time quantitative reverse transcription polymerase chain reaction and assessed its influence on murine melanoma cell functions, including migration. Cells treated with HCCP showed a higher inhibitory effect on migration compared to the control. According to the authors, HCCP (characterised by the increased expression of *A2M*) had significant anti-migration efficacy and may be considered a novel therapeutic agent for cancer treatment [34]. In both in vitro and in vivo studies, Lee et al. [35] showed that the downregulation of *A2M* and decreased A2M protein expression levels were positively correlated with decreased ADAM metallopeptidase with thrombospondin type 1 motif 1 (ADAMTS1). Moreover, bioinformatics analyses suggest that both A2M and ADAMTS1 expressions are correlated with more aggressive phenotypes of lung adenocarcinoma. Similar to our studies, low A2M expression was shown in adenocarcinoma tissues compared to healthy ones. A2M was shown to play a role in activating the epithelial–mesenchymal transition (EMT) and promoting metastasis in lung adenocarcinoma. A2M expression was negatively correlated with patients’ overall survival [35]. Zhang and collaborators described *A2M* downregulation in intrahepatic cholangiocarcinoma (ICC) compared to normal bile duct tissue, suggesting this gene’s potential utility as an adverse prognostic factor [25]. Moreover, the authors suggested that the positive relation between A2M overexpression and better prognosis for patients suffering from ICC indicated that A2M may be a potential drug used for cancer treatment [25]. Our first attempt at inhibiting canine OSA cell migration by adding A2M both at 10 mM and 30 mM (Figure 4) is the first promising result for A2M’s future use in veterinary oncology.

On the other hand, in human primary and metastatic OSA, *A2M* was shown to be upregulated [36,37]. Liu et al. suggested that *A2M* may act as oncogene and be consistently involved in the pathophysiological process of human osteosaroma [36], suggesting that, together with LRP1, it may be considered as potential therapeutic targets [37].

To the best of our knowledge, the present study is the first to establish lower A2M expression in canine OSA cells compared to normal osteoblasts and demonstrate its possible anti-migration capacity in wound-healing assays. The significant (*p* ≤ 0.05 or *p* ≤ 0.001) inhibition of canine OSA cell migration by adding either 10 mM or 30 mM A2M for both 12 h and 24 h cell culture demonstrates A2M’s potential as a therapeutic agent for metastatic canine OSA. However, A2M’s influence on the metastatic process of canine OSA, including the possible molecular mechanisms of inhibiting cancer cell migration, invasion, and proliferation, as well as its potential role in future antimetastatic therapies, require further in vitro and in vivo investigations. Further investigations should include the following: multidirectional studies on the molecular mechanism of A2M’s inhibition of migration, invasiveness and tumorgenesis, which, according to the current knowledge, is probably related to upregulation of LRP1 and frizzled receptor (FZD); the inhibition of the expression of Wnt ligands and, hence, the autocrine activation of Wnt/β-catenin signalling; and the relocation of beta cathenin and the induction of cellular cadherins, which act as tumor supressors [30]. There is also evidence that A2M regulates tumor cell growth by upregulating PTEN and inhibiting tumour-promoting signalling pathways such as PI3K/AKT and SMAD, making A2M a promising anticancer drug which should also be further assessed [25].

## 4. Materials and Methods

### 4.1. Cell Lines

Two canine OSA cell lines (OSCA-8 and OSCA-32) (Kerafast, Boston, MA, USA) and primary CnOb (Cell Applications, San Diego, CA, USA) were used in this study. The OSCA-8 cell line was derived from the left shoulder tumour of an intact 1-year-old male Rottweiler. The gene expression profile of this cell line was consistent with that established in the most malignant molecular phenotype of OSA [38,39]. The OSCA-32 cell line was derived from a tumour in the left wrist of a 9-year-old spayed female Great Pyrenees. The gene expression profile of the OSCA-32 cell line was consistent with that established in the less aggressive molecular phenotype of OSA [38,40].

CnOb obtained from normal healthy canine bones were used as a negative control.

### 4.2. Cell Culture

Osteosarcoma cells (OSCA-8 and OSCA-32—passages 12 and 15) were cultivated under aseptic conditions in sterile chambers with laminar air flow using Dulbecco’s Modified Eagle Medium (DMEM, Gibco, Waltham, MA, USA) supplemented with 10% foetal bovine serum (FBS, Life Technologies, Gibco, Waltham, MA, USA) and antibiotics (Primocin—InvivoGen, Waltham, MA, USA, and penicillin–streptomycin 100× solution—HyClone^TM^, Marlborough, MA, USA) and 1% HEPES buffer (HyClone^TM^, USA), while CnOb (passage 7) were grown using Canine Osteoblast Growth Medium (Cell Applications, San Diego, CA, USA) supplemented with 10% FBS (Life Technologies, Gibco, Waltham, MA, USA) and antibiotics (Primocin—InvivoGen, Waltham, MA, USA) and penicillin–streptomycin 100× solution—HyClone^TM^, Marlborough, MA, USA). Cultures were grown in a CO_2_ incubator under standard conditions (37 °C, 5% CO_2_, 95% humidity). All experiments were performed when the cells, in the logarithmic growth phase, reached 75–80% confluence. Cell viability was assessed with trypan blue using an Invitrogen Countess II automatic cell counter (Thermo Fisher, Waltham, MA, USA).

### 4.3. 2DE Electrophoresis and MALDI-TOF/TOF MS Analysis

#### 4.3.1. Protein Isolation, Cleaning, and Precipitation

The homogenisation and isolation of the proteins contained in cell pellets (OSCA-8, OSCA-32, CnOb, *n* = 7 per group) were performed using a solution containing RIPA buffer (Sigma-Aldrich, Saint Louis, MO, USA) and a protease inhibitor cocktail (Sigma-Aldrich, Saint Louis, MO, USA). In detail, the cell suspension in the test tube was centrifuged (300× *g*, 3 min, room temperature (RT)) and resuspended in 1.5 mL of phosphate-buffered saline (PBS) (Gibco, Waltham, MA, USA). A suspension originating from one culture flask was divided into three samples of 0.5 mL and centrifuged (300× *g*, 3 min, RT). Protease Inhibitor Cocktail and RIPA buffer were mixed in a proportion of 1:100. The supernatant was removed and 200 µL of RIPA buffer/protease inhibitor cocktail solution was added. After being vortexed (3×, 5 s every 5 min), the solutions were centrifuged (16,400× *g*, 15 min, 4 °C). A spectrophotometric method was used to determine the protein concentration using Maestro Nano (Maestrogen, Xinzhu, Taiwan) equipment, which measured the absorbance at 280 nm. Measurements were performed according to the manufacturer by dropping two microliters and transmitting an optical beam by passing a beam of electromagnetic radiation towards the blank solution.

#### 4.3.2. Two-Dimensional Gel Electrophoresis (2DE)

For the first dimension of electrophoresis, isoelectric focusing was performed. Pellets containing 200 µg of protein were dissolved in a rehydration buffer (Bio-Rad, Warsaw, Poland). The obtained protein solutions were applied to a rehydration plate (Hoefer IEF100, Hoefer, Inc., Holliston, MA, USA) and covered with 17 cm of immobilised pH gradient (IPG) linear strips for isoelectric focusing (pH 3–10, Bio-Rad, Warsaw, Poland). After 12 h of rehydration, strips with soaked proteins were transferred to an IEF-100 Hoefer apparatus (Hoefer IEF100, Hoefer, Inc., Holliston, MA, USA) for focusing. Afterwards, strips were equilibrated in dithiothreitol (1.30 mM) and iodoacetamide (1.35 mM) solutions.

For the second dimension, gel electrophoresis was performed with a 12.5% polyacrylamide gel using a Bio-Rad PROTEAN II xi Cell chamber. The conditions for vertical separation were as follows: 600 V, 30 mA, 30 W per gel. The electrophoretic chamber was filled with 0.025 M Tris/glycine buffer pH 8.3. Following protein electrophoretic separation, proteins were silver-stained, with silver nitrate used for impregnation along with formaldehyde for reduction, and the proteins were then scanned using an Image Scanner III (GE Healthcare, Chicago, IL, USA). The gel images were analysed in Delta2D (DECODON, Greifswald, Germany) software for statistical (one-way ANOVA, *p* ≤ 0.05) and graphical analyses, including manual selection of false positive and false negative spots. The selected protein spots were removed from the polyacrylamide gel with a scalpel, transferred to microtubes, rinsed with water, and discoloured. Reduction and alkylation were performed with dithiothreitol and iodoacetamide, respectively. The gel pieces were digested by trypsin according to the manufacturer’s instructions (Trypsin Gold, Promega, Madison, WI, USA) in 50 mM of an ammonium bicarbonate water solution environment for 12 h in an incubator (37 °C). Proteins were extracted from gels with a 50:45:5 acetonitrile solution (Merck, Rahway, NJ, USA), water, and trifluoroacetone acid (Merck, Rahway, NJ, USA) using an ultrasonic bath at room temperature. The extraction was repeated three times, and every step lasted 15 min. Finally, extracts were concentrated in a CentriVap (Labconco local seller A.G.A Analytical, Warsaw, Poland). The obtained peptide pellets were dissolved in 10 µL of 0.1% trifluoroacetone acid and purified using Sample Prep Pipette Tips (ZipTip 0.2 μL, C18, Millipore, Merck, Rahway, NJ, USA).

#### 4.3.3. MALDI-TOF/TOF MS Analysis

We performed a MALDI-TOF/TOF MS analysis by applying 1 µL of a purified protein sample to an AnchorChip detection plate. Then, when the sample was completely dry, 1 µL of the α-cyano-4-hydroxycinnamic acid (Bruker, Bremen, Germany) matrix was used for covering sample spots. At the same time, 0.5 µL of a peptide standard was spotted on the calibration field (Peptide Calibration Standard II, Bruker, Bremen, Germany). Mass spectra were recorded in the range of 700–4000 Da using a MALDI-TOF/TOF spectrometer (Ultraflextreme MALDI TOF/TOF, Bruker, Bremen, Germany) and flex Control 3.3 software (Bruker, Bremen, Germany). Next, using flex Analysis 3.0 software (Bruker, Bremen, Germany), a list of peaks with signal-to-noise ratios >3 was created. After removing environmental contaminants, the peak list was transferred to BioTools 3.2 (Bruker, Bremen, Germany) and compared to Mascot 2.2 using a Swiss-Prot database [https://www.uniprot.org/, accessed on 20 December 2020]. The results of the MALDI-TOF/TOF MS analysis were assessed using the following criteria from previously established protocols [10,11]: *p* ≤ 0.05, Mascot Score ≥ 61, Rt OSCA-8/CnOb and OSCA-32/CnOb ≤0.67 or ≥1.5, according to previously published protocols.

### 4.4. Validation of Protein Expression Using Simple Western

#### 4.4.1. Protein Isolation, Cleaning, and Precipitation

Homogenisation and isolation of the protein contained in the cell pellets (OSCA-8, OSCA-32, CnOb, *n* = 3 per group) were performed in the same manner as described in Section 4.3.1. The homogenates were analysed for protein concentration using the bicinchoninic acid assay (BCA) method. The BCA reagent was prepared according to the manufacturer’s protocol for the BCA Protein Assay Kit (Abcam, Cambridge, UK). On a 96-well plate, 200 µL of the BCA reagent and 10 µL of protein homogenate were applied to each well (three technical replicates for each sample). To prepare the standard curve, we used a starting concentration of 1 mg/mL of bovine serum albumin (BSA, Sigma-Aldrich). In the standard curve, the following concentrations of BSA/RIPA solution were used: 0, 200×, 400×, 600×, 800×, and 1000×. A spectrophotometric analysis was performed at a temperature of 37 °C and wavelength of 562 nm using an Infinite 200 PRO M Nano Plate Reader (Tecan, Zurich, Switzerland) with Tecan I-Control 2.0 Software (Tecan, Zurich, Switzerland).

#### 4.4.2. Qualitative Protein Expression Analyses

A2M expression detected with MALDI TOF/TOF MS analysis was validated by the Simple Western technique using a Jess analyser (Protein Simple, Minneapolis, MN, USA). Standard pack reagents were prepared according to Jess guidelines with reagents from the EZ Standard Pack (Bio-Techne, Minneapolis, MN, USA). The EZ Standard Pack 12–230 kDa (Bio-Techne, Minneapolis, MN, USA) contains a ready-to-use biotinylated ladder, a fluorescent 5× master mix, and a DTT solution. In a clear tube, DTT was mixed with 40 µL of deionised water to produce a 400 mM solution. The master mix was mixed with 20 µL of sample buffer and 20 µL of prepared 400 mM DTT solution. The optimal protein concentration depends on the expression level of the assessed protein. The protein homogenates were mixed with a sample diluent: 0.1× sample buffer (Bio-Techne, Minneapolis, MN, USA). One part of the 5× fluorescent master mix was combined with four parts of diluted lysate in a microcentrifuge tube. Denaturation of the samples was performed at 95 °C for 5 min. In the next step, the samples were vortexed, spun for 10 s and at 3000 rpm/min and then stored on ice. Primary antibody anti-human alpha-2-macroglobulin MAB1938 (Bio-Techne, Minneapolis, MN, USA) was prepared and diluted with Antibody Diluent 2 (Bio-Techne, Minneapolis, MN, USA). A luminol-S and peroxide mix (in 1:1 proportion) was vortexed and stored on ice. All of the reagents, including a secondary anti-mouse antibody (Bio-Techne, Minneapolis, MN, USA), were applied to the 12–230 kDa separation module according to manufacturer guidelines.

##### Linear Range of Proteins and Primary Antibody Saturation Assessment

Before performing the main part of the experiment, the linear range of the proteins and 90% saturation of the primary antibody were determined to ensure the proper antibody and sample concentrations for A2M expression analyses. The following protein concentrations were used in each sample to assess the linear range of the proteins: 0.1; 0.2; 0.4; 0.5; and 0.6 mg/mL.

The concentration of anti A2M antibody MAB1938 (Bio-Techne, Minneapolis, MN, USA) was used in the following antibody dilutions for antibody saturation assessment: 1:6.25; 1:12.5; 1:50; 1:100; 1:200; and 1:400.

### 4.5. Migration Rate Assessment in Canine OSA Cell Lines Treated with A2M

Cells were seeded in culture inserts (2-well Ibidi culture inserts, Ibidi GmbH, Grafelfing, Germany) at a density of 3 × 10^4^ cells in each well. When around 90% confluence was reached, the culture inserts were removed. In the next step, the medium was removed, and mitomycin C (Abcam, Cambridge, UK) was added to a final concentration of 10 μg/mL. Cell monolayers were incubated at 37 °C under standard conditions. After 3 h of incubation, the medium with mitomycin C was replaced with a serum-free medium in the control group, and an A2M-enriched medium was used in the second and third groups, with concentrations of 10 mM and 30 mM, respectively. Mitomycin C is a DNA synthesis inhibitor used for cell migration assessment. It ensures minimal loss of viability and maximizes the inhibition of cell division. Cells pre-treated with MMC inhibit cell proliferation, eliminating the contribution of those cells to wound closure and enabling cancer cell migration to truly be assessed [41,42]. To detect cancer cell migration, in the case of adherent cells, such as in our study, the cells need to bind to the bottom of the wells, and proliferation needs to be inhibited so that migration can be assessed [41]. Images of cells migrating within the scratch were captured at specific time points (t0 = 0, t1 = 12 h, t2 = 24 h) using an inverted microscope (Primovert, Zeiss, Munich, Germany) at 4× magnification. The captured figures were analyzed afterwards using Zen Pro 2012 (Zeiss, Munich, Germany), and cancer cell migration was evaluated by calculating the distance between the edges of the scratches (100 measurements for each scratch) according to the procedure described by Rodriguez et al. [43]. The experiment was repeated in triplicate. The wound-healing assay was also used to compare the migration rate of the canine OSA cell lines OSCA-8 and OSCA-32 to the migration rate of the CnOb. The experiment was performed in the same manner as described above, but in a serum-free medium for all groups.

## 5. Conclusions

Using MALDI-TOF/TOF MS analysis, eight proteins (GTF2I, MIPOL-1, CFAP298, PGAM1, UB2L6, EDARADD, LRRC72, and A2M) were found to be significantly differentially (*p* ≤ 0.05) expressed in canine OSA cell lines (both OSCA-8 and OSCA-32) compared to CnOb. CFAP298 and GTF2I were upregulated, whereas MIPOL-1, PGAM1, UB2L6, EDARADD, LRRC72, and A2M were downregulated in OSA cells. The downregulated expression of A2M proteins was subsequently confirmed with a Simple Western qualitative analysis. The wound-healing assay results revealed a positive correlation between the malignant phenotype of OSA cells and the migration rate of canine OSA cells within the first 12 h. Moreover, we established a negative correlation between the migration rate of canine OSA cells treated with A2M, demonstrating that A2M acts as a migration inhibitor of canine OSA cells. This study may be the first report indicating A2M’s possible use in canine OSA cell metastasis treatment. However, further in vitro and in vivo studies are needed to confirm whether A2M can also act as an anti-metastatic agent in this malignancy.

## Figures and Tables

**Figure 1 ijms-25-03989-f001:**
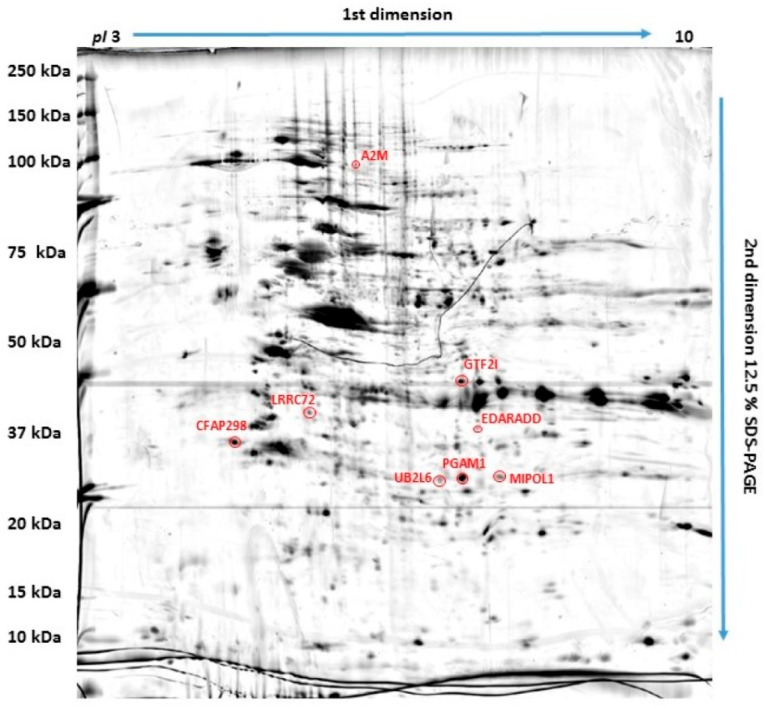
Polyacrylamide gel with eight red-marked protein spots which were selected as significantly differentially expressed in osteosarcoma cells compared to canine osteoblasts.

**Figure 2 ijms-25-03989-f002:**
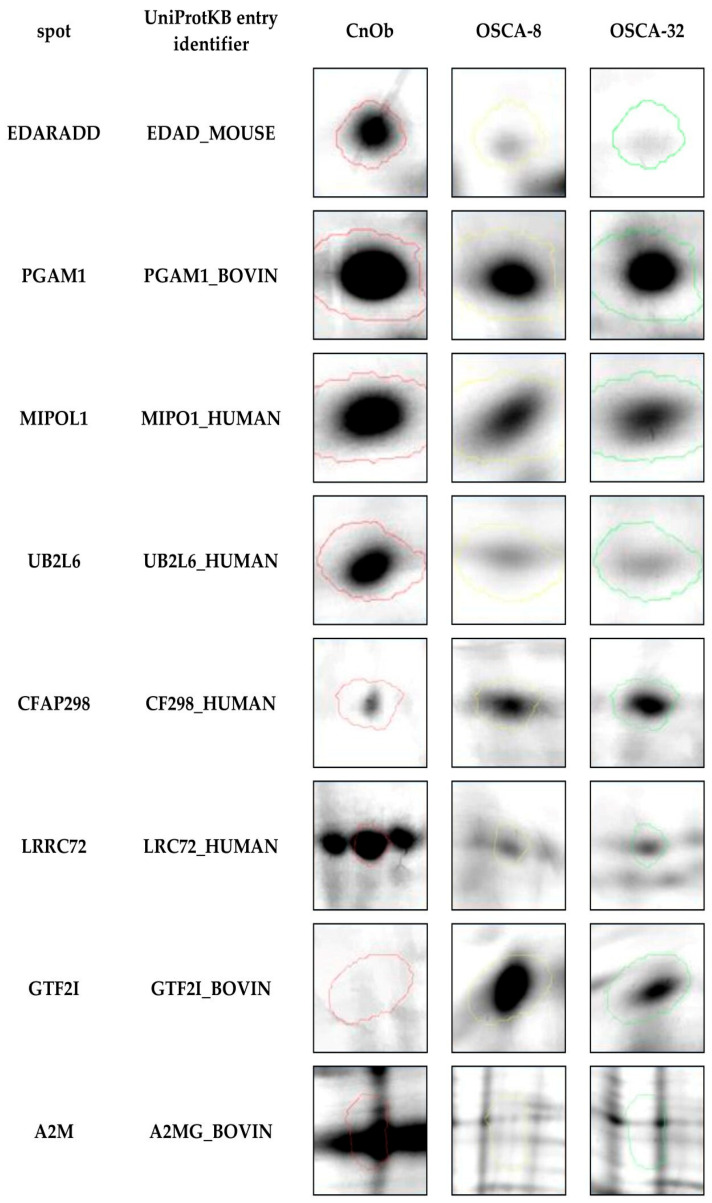
Representative two-dimensional electrophoresis gels of significantly (*p* ≤ 0.05) differentially expressed proteins in canine osteosarcoma cells (from the OSCA-8 and OSCA-32 cell lines) versus canine osteoblasts (CnOb).

**Figure 3 ijms-25-03989-f003:**
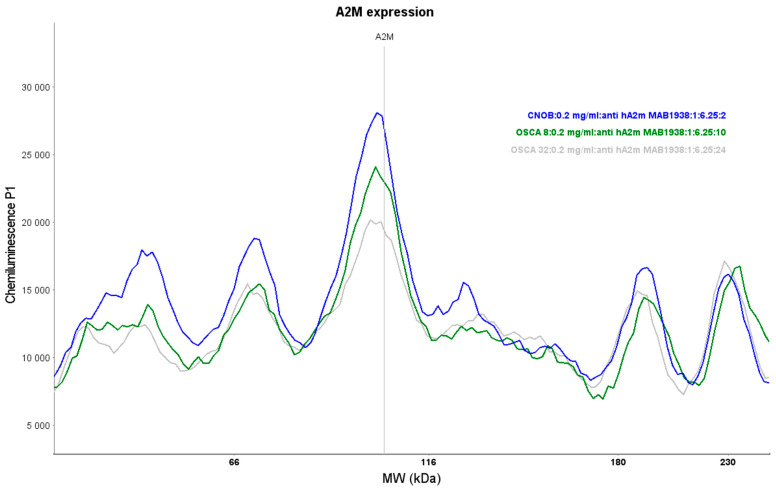
The Simple Western assessment of A2M expression in CnOb (blue line) as well as the OSCA-8 (green line) and OSCA-32 (grey line) cell lines.

**Figure 4 ijms-25-03989-f004:**
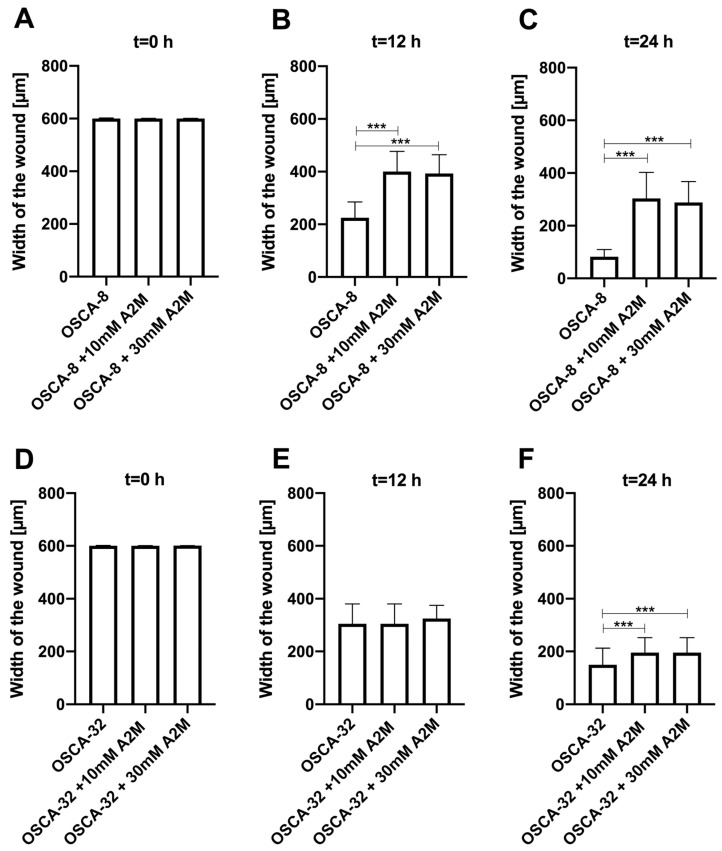
Bar graphs showing the A2M protein’s effect in two concentrations (10 mM and 30 mM) on inhibiting the migration of canine osteosarcoma cells from the OSCA-8 cell line at (**A**)—t0 (0 h); (**B**)—t1 (12 h); (**C**)—t2 (24 h) and OSCA-32 cell line at (**D**)—t0 (0 h); (**E**)—t1 (12 h); (**F**)—t2 (24 h); ***—high statistical significance at *p* ≤ 0.001.

**Figure 5 ijms-25-03989-f005:**
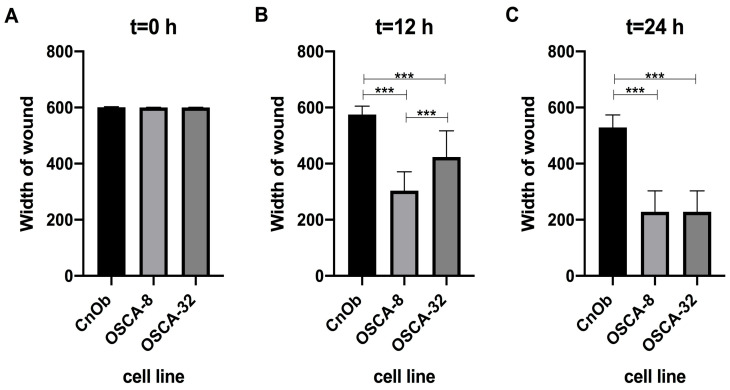
Bar graphs showing the migration rate of canine OSCA-8 and OSCA-32 compared to the control cell line CnOb at (**A**)—t0 (0 h); (**B**)—t1 (12 h); and (**C**)—t2 (24 h). ***—high statistical significance at *p* ≤ 0.001.

**Figure 6 ijms-25-03989-f006:**
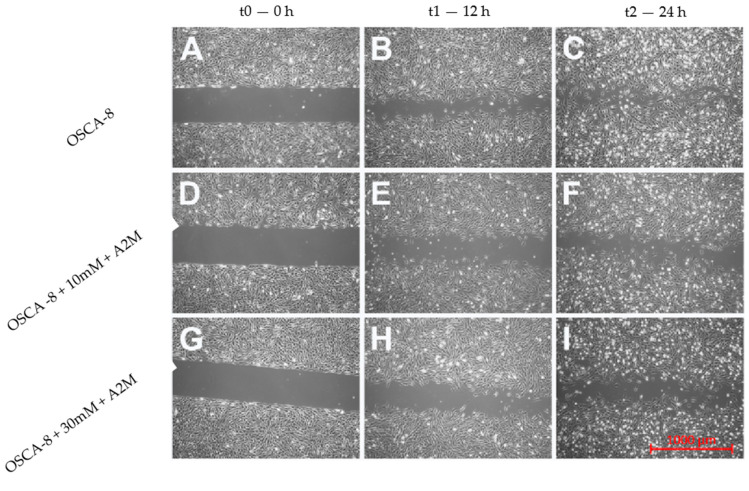
Wound-healing assay microscopic images showing the inhibition of canine OSCA-8 osteosarcoma cell migration by A2M (scale bar). Untreated cells at t0 (0 h) (**A**); t1 (12 h) (**B**); t2 (24 h) (**C**); cells incubated with 10 Mm A2M at t0 (0 h) (**D**); t1 (12 h) (**E**); t2 (24 h) (**F**); cells incubated with 30 mM A2M at t0 (0 h) (**G**); t1 (12 h) (**H**); t2 (24 h) (**I**).

**Figure 7 ijms-25-03989-f007:**
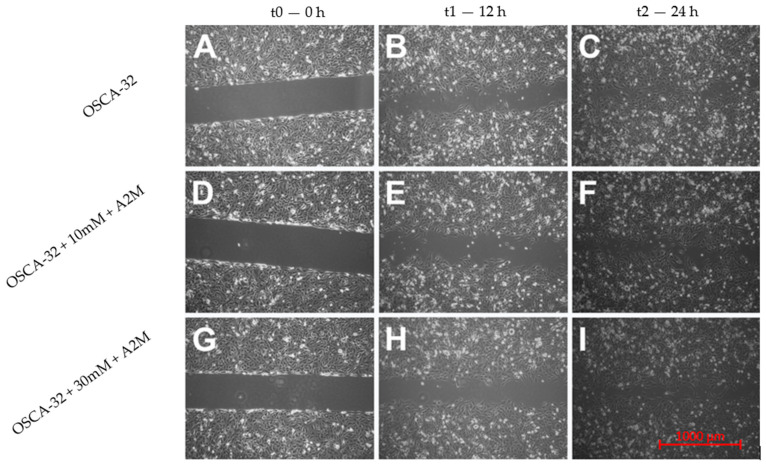
Wound-healing assay microscopic images showing the inhibition of canine OSCA-32 osteosarcoma cell migration by A2M. Untreated cells at t0 (0 h) (**A**); t1 (12 h) (**B**); t2 (24 h) (**C**); cells incubated with 10 Mm A2M at t0 (0 h) (**D**); t1 (12 h) (**E**); t2 (24 h) (**F**); cells incubated with 30 mM A2M at t0 (0 h) (**G**); t1 (12 h) (**H**); t2 (24 h) (**I**).

**Table 1 ijms-25-03989-t001:** Significantly (*p* ≤ 0.05) differentially expressed proteins in canine osteosarcoma cells (for OSCA-8 and OSCA-32 cell lines) versus canine osteoblasts (CnOb) identified using MALDI-TOF/TOF MS.

ID	Protein	Accession Number (UniProtKB)	Score	Match	MW (Da) **	pI *	Modif.	Seq. Cov (%)	Rt CnOb/OSCA-32	Rt CnOb/OSCA-8
1	Ubiquitin/ISG15-conjugating enzyme E2	O14933	95	6	17,757	7.71	C, Ac, Ox	26	0.664	0.383
2	Leucine-rich-repeat-containing protein 72	A6NJI9	150	14	33,863	8.91	C, Ac, Ox	36	0.471	0.287
3	General transcription factor II-I	A7MB80	84	13	110,569	7.13	C, Ac, Ox	14	4.14	8.65
4	Cilia- and flagella-associated protein 298	P57076	68	10	33,374	6.99	C, Ac	27	3.78	6.46
5	Mirror-image polydactyly 1 protein	Q8TD10	105	11	51,847	5.55	C, Ac, Ox	17	0.56	0.60
6	Alpha-2-macroglobulin	Q7SIH1	158	14	168,953	5.71	C, Ac, Ox	17	0.19	0.48
7	Ectodysplasin-A receptor-associated adapter protein	Q8VHX2	74	5	24,251	5.04	C, Ac, Ox	33	0.59	0.23
8	Phosphoglycerate mutase 1	Q3SZ62	77	10	28,852	6.67	C, Ac, Ox	37	0.71	0.64

Abbreviations: C—carbamidomethylation of cysteine; Ox—oxidation of methionine; Ac—acetylation of protein; N-term. Listed molecular weights and pI values correspond to the MASCOT search results. * Protein score is −10 × Log(P), where P is the probability that the observed match is a random event. For mammals taxonomy, protein scores greater than 61 are significant (*p* < 0.05). ** Listed molecular weights and pI (isoelectric point) values correspond to the MASCOT Search Result.

## Data Availability

On request, raw data obtained within this study can be obtained from the Authors (S.S.W., K.M. and K.Z.-K.).

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
