# Peer review of "Proteomic Analyses Reveal the Role of Alpha-2-Macroglobulin in Canine Osteosarcoma Cell Migration"

_ijms, 2024, doi:10.3390/ijms25073989_

Round 1

Reviewer 1 Report

Comments and Suggestions for Authors

Dear Editor and Dear Authros,

I read the masnucript and I found it very interesting. It is well written and easy to understand.

The research aims to investigate the biological mechanisms of OSA in order to go deeper inside this pathology and to try to hyphotethise new therapeutic strategies. The manuscript is well written because authors explain clearly the problem of this pathology (both in human and veterinary medicine) and highlights that a few therapeutic strategies are available. This limitation is due not only to the few molecules active against OSA but also to some mechanisms resistance. This research tries to fill this gap in therapeutics. This research proposed the results obtain by a complex series of different experiments that have never been exposed according to OSA.  The methodologies proposed are in part quite common (for example, 2D SDS page) or innovative, such as MALDI TOF, that are usulally used to identify bacteria. According to this, a few manuscript are dealing with the application of this methodology to evaluate and study neoplastic cells. The conclusion is consistent with the entire manuscript. Figures and tables support the concept presented along the manuscript.

Here a provide a short list of comments, requiring minor corrections:

L63: add reference number after “et al “

L67: add reference number after “et al “

L71-72: maybe this sentences would sound better at the beginning of conclusion section

In some parts along the introduction section, the text seems to be grey and not black. Please, just check it with the Editorial office during the proof-check.

L213: also the cost to purchase the machine is not negligible… and qualified personnel is required. I think that it is better to add these two items at the list of cons

Reviewer 2 Report

Comments and Suggestions for Authors

Dear Authors,

The author needs to address these comments will further enhance the manuscript's clarity, rigor, and overall quality.

1.      The author need to proofread the manuscript for language and grammar to enhance overall clarity. Consider revising some sentences for better readability.

2.      P10, Line 219 – Check for spelling mistakes.

3.      P14, Line 426 – Please provide rationale for using mitomycin C to inhibit cell proliferation and ensure that cell migration is the primary readout in the wound healing assay.

4.      P13, Line 394 – Author need to provide specific molecular mechanisms underlying A2M inhibitory effects on cancer cell migration, invasion and proliferation.

5.      P11, Line 288 – Author need to provide additional details regarding passage number of the cell lines used.

Comments on the Quality of English Language

Minor english corrections and spelling erros

Round 2

Reviewer 2 Report

Comments and Suggestions for Authors

Hi,